# Volume of hyperintense inflammation ($V_{HI}$): A quantitative imaging biomarker of inflammation load in spondyloarthritis, enabled by human-machine cooperation

Carolyna Hepburn[1,2], Alexis Jones[1,3], Alan Bainbridge[4], Coziana Ciurtin[3], Juan Eugenio Iglesias[2,5,6], Hui Zhang[2‡], Margaret A. Hall-Craggs[1,7‡], Timothy J. P. Bray[1,7‡]*

**1** Centre for Medical Imaging, University College London, London, United Kingdom, **2** Centre for Medical Image Computing, University College London, London, United Kingdom, **3** Centre for Adolescent Rheumatology, University College London, London, United Kingdom, **4** Department of Medical Physics, University College London Hospitals, London, United Kingdom, **5** Computer Science and Artificial Intelligence Laboratory, Massachusetts Institute of Technology, Boston, Massachusetts, United States of America, **6** Martinos Center for Biomedical Imaging, Massachusetts General Hospital and Harvard Medical School, Boston, Massachusetts, United States of America, **7** Imaging Department, University College London Hospital, London, United Kingdom

‡ HZ, MAHC and TJPB are joint senior authorship on this work.

* t.bray@ucl.ac.uk

**Data Availability Statement:** The data, code and models are available at https://github.com/c-hepburn/Bone_MRI.

## Abstract

Qualitative visual assessment of MRI scans is a key mechanism by which inflammation is assessed in clinical practice. For example, in axial spondyloarthritis (axSpA), visual assessment focuses on the identification of regions with increased signal in the bone marrow, known as bone marrow oedema (BMO), on water-sensitive images. The identification of BMO has an important role in the diagnosis, quantification and monitoring of disease in axSpA. However, BMO evaluation depends heavily on the experience and expertise of the image reader, creating substantial imprecision. Deep learning-based segmentation is a natural approach to addressing this imprecision, but purely automated solutions require large training sets that are not currently available, and deep learning solutions with limited data may not be sufficiently trustworthy for use in clinical practice. To address this, we propose a workflow for inflammation segmentation incorporating both deep learning and human input. With this 'human-machine cooperation' workflow, a preliminary segmentation is generated automatically by deep learning; a human reader then 'cleans' the segmentation by removing extraneous segmented voxels. The final cleaned segmentation defines the volume of hyperintense inflammation ($V_{HI}$), which is proposed as a quantitative imaging biomarker (QIB) of inflammation load in axSpA. We implemented and evaluated the proposed human-machine workflow in a cohort of 29 patients with axSpA who had undergone prospective MRI scans before and after starting biologic therapy. The performance of the workflow was compared against purely visual assessment in terms of inter-observer/inter-method segmentation overlap, inter-observer agreement and assessment of response to biologic therapy. The human-machine workflow showed superior inter-observer segmentation overlap than purely manual segmentation (Dice score 0.84 versus 0.56). $V_{HI}$ measurements produced by the

**Funding:** This work was funded by Action Medical Research, the Humanimal Trust and The Albert Gubay Foundation. TJPB was supported by an NIHR Clinical Lectureship (CL-2019-18-001). TJPB and MHC are supported by the National Institute for Health Research (NIHR) Biomedical Research Centre (BRC). This work was undertaken at UCLH/UCL, which receives funding from the UK Department of Health's the NIHR BRC funding scheme. The views expressed in this publication are those of the authors and not necessarily those of the UK Department of Health. The funders had no role in study design, data collection and analysis, decision to publish, or preparation of the manuscript.

**Competing interests:** The authors have declared that no competing interests exist.

workflow showed similar or better inter-observer agreement than visual scoring, with similar response assessments. We conclude that the proposed human-machine workflow offers a mechanism to improve the consistency of inflammation assessment, and that $V_{HI}$ could be a valuable QIB of inflammation load in axSpA, as well as offering an exemplar of human-machine cooperation more broadly.

# 1 Introduction

Qualitative assessment of MRI scans is the main mechanism by which inflammation, a complex biological response to harmful stimuli, is assessed in clinical practice. For example, in spondyloarthritis, areas of increased ('hyperintense') signal in the subchondral bone on water-sensitive images (such as those generated by the widely-used short tau inversion recovery (STIR) sequence) are referred to as bone marrow oedema (BMO) and form part of the diagnostic criteria in this disease [1]. The extent of BMO (in terms of the number of involved quadrants as well as the presence of particularly bright or deep areas of inflammation) is also the key feature used in the Spondyloarthritis Research Consortium of Canada (SPARCC) system for qualitative assessment of the burden of inflammation [2], although this is a research tool (typically required substantial reader expertise and often also calibration exercises) which is not used in clinical practice. In standard clinical care, despite the important role that STIR MRI plays in diagnosis, quantification and monitoring of inflammation, images are typically interpreted in a purely qualitative fashion. This introduces a source of subjectivity and consequently evaluation of inflammation burden can vary widely depending on reader expertise and the clinical setting.

The size of the problem was highlighted by a 2017 survey of 269 radiologists, which found wide variation in the use of MRI for assessing spondyloarthritis, including imaging sequences, anatomical coverage and image interpretation [3]. Only 75% of radiologists reported awareness of spondyloarthritis as a disease entity, whilst only 25–31% were aware of formal MRI definitions of inflammation [3]. Even in a controlled research setting, there is wide disparity in readers' agreement on the presence and severity of inflammation [4–7]. This inconsistency creates a major risk of misinterpretation/misdiagnosis and inappropriate treatment. In clinical trials, it contributes to reduced power/increased sample size and increased cost.

In addition to the difficulties with interpretation, clinical radiological reports in spondyloarthritis are descriptive without quantitative assessment of inflammation. This introduces scope for misunderstanding of the report, particularly when styles differ between radiologists [8]. A quantitative, easily-understandable biomarker of inflammation could potentially simplify interpretation substantially for the recipients of these reports.

Deep learning-based segmentation is a natural approach to segmenting and quantifying inflammation, and could help to improve the objectivity of inflammation assessment. However, purely automated segmentation algorithms can require large, carefully curated and labelled training sets that are not currently available for this field (and many others) in order to reach sufficient performance for use in a clinical workflow. Furthermore, there is an evolving discussion around the importance of 'trustworthiness' of artificial intelligence in medical imaging [9]. Trustworthiness has a number of facets including the transparency and explainability of the component algorithms [9]. A workflow incorporating discrete algorithms, each performing relatively simple tasks, that allows radiologists to make a final judgement about the presence of inflammation might better satisfy these criteria than more complex algorithms

attempting to perform multiple tasks in one step without human intervention. This may be particularly important when the size of the training dataset available is limited.

In light of these considerations, we propose a hybrid 'human-machine' workflow for inflammation quantification, aiming to combine deep learning-based segmentation with human control and oversight of the image assessment. The final segmentation from this human-machine workflow defines the volume of hyperintense inflammation ($V_{HI}$), which we propose as a quantitative imaging biomarker of inflammation load. We hypothesise that this biomarker can provide an accurate, precise and responsive method of scoring inflammation for use in clinical practice.

## 2 Materials and methods

### 2.1 Overview of study design

We aimed to develop a hybrid 'human-machine' segmentation workflow for measuring the volume of hyperintense inflammation ($V_{HI}$), which is proposed as a quantitative imaging bio-marker of inflammation load. To do this, we implemented and evaluated this workflow in a prospectively-acquired dataset. To assess the performance of $V_{HI}$ as a biomarker, we assessed its relationship with visual scoring, inter-observer agreement and responsiveness to biologic therapy in patients with spondyloarthritis who underwent scans before and after biologic therapy. The data, code and models used in the study are available at https://github.com/c-hepburn/Bone_MRI.

### 2.2 Study cohort

This study was performed with institutional review board approval (REC reference 15/LO/1475), and all subjects gave written informed consent.

Data were taken from a completed prospective longitudinal study conducted at *(anonymised)* hospital between April 2018 and July 2019 (29 subjects consisting of 13 males, 16 females; mean age 42.4 years) with the aim of evaluating the ability of quantitative imaging bio-markers to measure and predict response to biologic therapy.

Potential participants were identified from clinical records of patients due to start biologic therapy and were initially approached about participation by rheumatologists at UCLH. Patients were included in the study if they were aged 18 to 85 years with a diagnosis of axial spondyloarthritis according to 2009 Assessment of SpondyloArthritis international Society (ASAS) criteria [1] and active disease according to the National Institute of Clinical Excellence (NICE guidelines NG65) criteria. Exclusion criteria included contraindications to MRI such as metallic implants, pacemaker, severe claustrophobia, pregnancy, body weight > 150kg, treatment with an oral, intra-articular or intra-muscular glucocorticoid within 4 weeks. All patients underwent MRI scan of the sacroiliac joints, and continued in the study if their MRI fulfilled ASAS criteria for sacroiliitis [10] and were eligible for their first biologic drug (biologic naive) or a change biologic therapy (switchers) in accordance with best practice (NICE guidelines NG65). A repeat scan was performed after 12 weeks (+/- 2 weeks) of continuous anti-tumour necrosis factor (anti-TNF) treatment or 16 weeks (+/- 2 weeks) of anti-interleukin-17 (IL-17) treatment. Patients were withdrawn from the study if biologic therapy was declined, contraindicated or stopped owing to adverse events.

### 2.3 Clinical assessments

Information regarding patient demographics (age, sex and ethnicity), disease duration, history of peripheral arthritis and enthesitis, extra-articular manifestations, human leucocyte antigen

(HLA) B27 status, drug history and smoking history were recorded at baseline. Symptom scores, comprising the Bath Ankylosing Spondylitis Disease Activity Index (BASDAI) and Ankylosing Spondylitis Disease Activity Score (ASDAS) as well as C-reactive protein (CRP) and erythrocyte sedimentation rate (ESR) were recorded at baseline and after 12–16 weeks of continuous treatment. A clinical response was assessed on the basis of a BASDAI improvement of $\geq 1.2$ and an improvement in spinal visual analogue score (VAS) of $\geq 1$. This criterion is in accordance with NICE criteria and was chosen to reflect real-world clinical practice in the UK. Other clinical response measures included a reduction in BASDAI by 50% (BASDAI 50), a clinical important improvement in ASDAS (CII ASDAS) defined as a change in ASDAS $>1.1$ and inactive disease defined as an ASDAS of $< 1.3$ (ASDAS ID).

## 2.4 Image acquisition

Images were acquired on a 3T Philips Ingenia scanner. Both conventional and quantitative MRI scans were acquired for the study. Here, we focused on the conventional MRI protocol data to ensure wide applicability, although the workflow is general and could also be applied to quantitative MRI data. The conventional MRI protocol consisted of STIR and T1-weighted turbo spin echo sequences acquired in an oblique coronal plane (parallel to the sacrum) with fixed field of view. Quantitative MRI sequences, consisting of Dixon and diffusion-weighted MRI, were also used but not analysed for the present study. For the STIR acquisition, parameters included: TR 5316ms, TE 50ms, TI 210ms, echo train length 21, slice thickness 3mm, pixel spacing 0.59x0.59mm, image matrix 336x336, number of slices 23–25. All data were anonymised prior to export from the scanner and subjects were given unique study identifiers for data handling.

## 2.5 Deep learning-enabled segmentation workflow

**2.5.1 Workflow overview.** Inflammation was segmented using a hybrid 'human-machine' workflow incorporating deep learning as well as human interpretation; a schematic illustration is shown in Fig 1. Rather than training a neural network on areas of disease, the network is simply trained to recognize potentially-inflamed areas of bone, which are referred to as 'disease regions', and then a threshold is used within these disease regions to segment inflammation. This approach has the advantages that (i) training a network to detect potentially-inflamed regions (rather than direct recognition of pathology) is a simple task which can be achieved with a relatively small dataset, (ii) the disease-region segmentations can easily be propagated onto images from other sequences, thus enabling assessment of disease with multiple sequences if desired, and (iii) the final threshold-based segmentation step is transparent and easily-understood.

The pipeline comprises the following steps:

i. *'Normal bone region' segmentation*
   Normal bone marrow in the interforaminal region of the sacrum (which is typically spared from inflammation) is segmented, using STIR images. Here, we performed this step manually to ensure that any artifacts or vessels were avoided, although this step can be straightforwardly automated using deep learning.

ii. *Estimation of STIR intensity threshold*
   The normal bone segmented in (i) is used to select a threshold towards the upper end of the normal bone intensity distribution, enabling separation of normal from inflamed marrow within the disease region (described below).

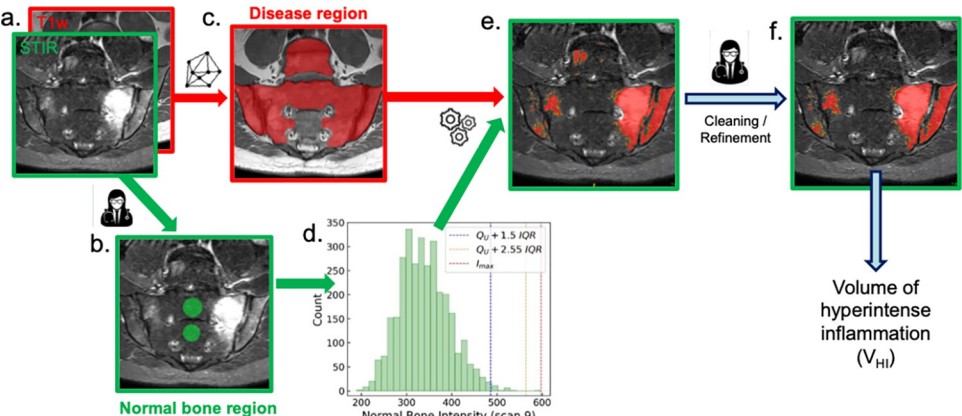

**Fig 1. Schematic illustration of the proposed 'human-machine' workflow.** A STIR image (outlined in green) and corresponding T1w image (outlined in red) (a) are used as the input. In this case, the STIR image shows left-sided sacroiliac joint inflammation. The **normal bone region** is manually defined by a radiologist to determine the distribution of intensity values of normal bone (b). Manual segmentation is used for this step to enable the radiologist to exercise judgement over the most representative area of normal marrow. The **disease region** (region of potential inflammation) is automatically segmented by a convolutional neural network to determine areas of potential disease (c). The normal bone intensity distribution is determined from the normal bone, and two thresholds are defined [the maximum of the intensity distribution (upper threshold, red dotted line) and a multiple of the interquartile range (lower threshold, orange dotted line); (a blue dotted line represents an empirical threshold value). Areas of tissue within the disease region above these thresholds are then denoted inflamed (e). Areas meeting the lower threshold alone are shown in yellow, whereas those meeting the upper threshold are shown in red. The final cleaning is performed by a radiologist to remove areas of artifact or inflammation outside the target area (f). In this example, areas of inflammation above the L5/S1 disc and in the sacroiliac joint space were removed (along with an artifactual lesion in the L5 vertebral body which was deemed to be due to a vessel) to facilitate a direct comparison with SPARCC visual scoring.

iii. *'Disease region' segmentation*

Areas of potential inflammation (all of the imaged bone in the pelvis, including bone adjacent to the SIJs, apart from the normal bone region) are segmented on T1W images using a supervised convolutional neural network with U-net architecture [11].

iv. *Thresholding within the disease region*

Voxels in the disease region are assigned labels of 0 if voxel signal intensity is below the intensity threshold and 1 if above the intensity threshold

v. *Automatic removal of very small segmented regions*

Regions containing <4 pixels (i.e. any region with an area <1.39mm$^2$), which are commonly due to noise or small vessels within the bone marrow, are automatically removed.

vi. *Manual correction of the final segmentation by a human observer*

Erroneous regions in the initial segmentation (e.g. areas of artefact or prominent vessels in the bone marrow) are removed by the radiologist. Once the correction procedure is complete, the final corrected segmentation defines the volume of STIR-hyperintense inflammation ($V_{HI}$), which is the proposed biomarker of inflammation load. $V_{HI}$ can be defined in terms of the volume *per se* (e.g. in mm$^3$) or as a voxel count (the former is simply the voxel count multiplied by the volume of an individual voxel).

**2.5.2 Details of step (ii)–Determining the segmentation threshold from the 'normal bone region'.**    Two different thresholds were obtained from the distribution of intensity values in the normal bone region, in order to provide one 'conservative' and one 'sensitive'

segmentation. To provide the 'conservative' segmentation, an upper limit, $L_{upper}$ was defined as the maximum intensity, $I_{max}$ of the distribution. To provide the 'sensitive' segmentation, a lower limit, $L_{lower}$ was computed as the sum of the upper quartile, $Q_U$ and a multiple, n of the inter-quartile range, IQR of the distribution $L_{lower} = QU + n \cdot IQR$. The multiple was determined automatically in order to adapt the 'sensitivity' for each scan for each patient: starting with the value of 1.5, the multiple n was incremented by 0.05 until the difference between upper and lower limits was less than the half of the interquartile range, $0 < L_{upper} - L_{lower} < IQR/2$. If the condition was initially satisfied, no incrementation was performed. Note that, for the primary analyses in this study, we used the lower, 'sensitive' threshold to determine $V_{HI}$.

**2.5.3 Details of step (iii)–Disease region segmentation.** Step (iii), i.e. the disease region segmentation, employed a convolutional neural network with 2D U-net architecture. To enable an assessment of generalisability, the network was first trained and tested on a subset of the full dataset (consisting of 248 T1-weighted image slices from 10 subjects), before a further evaluation of segmentation performance was performed by qualitative visual assessment on the remaining 19 subjects.

*Reference standard.* The reference standard for training was manual segmentation of the disease region, including all bone in the imaged pelvis, the sacroiliac and facet joint spaces (Fig 2). The segmentation was performed by a postdoctoral researcher with two years of MRI experience who had received training in interpretation of the relevant anatomy by a radiologist; this radiologist also performed a slice-by-slice review of the segmentations in a subset of the cases to ensure that these anatomical assessments were accurate. Segmentations were performed using ITK-SNAP Version 3.8 [12].

*Data partition.* Data was first partitioned at subject level at random into two sets: 200 image slices (8 subjects) for training with four-fold cross validation to find the optimal set of hyper-parameters and 48 image slices (2 subjects) for testing. The first set was then subdivided four

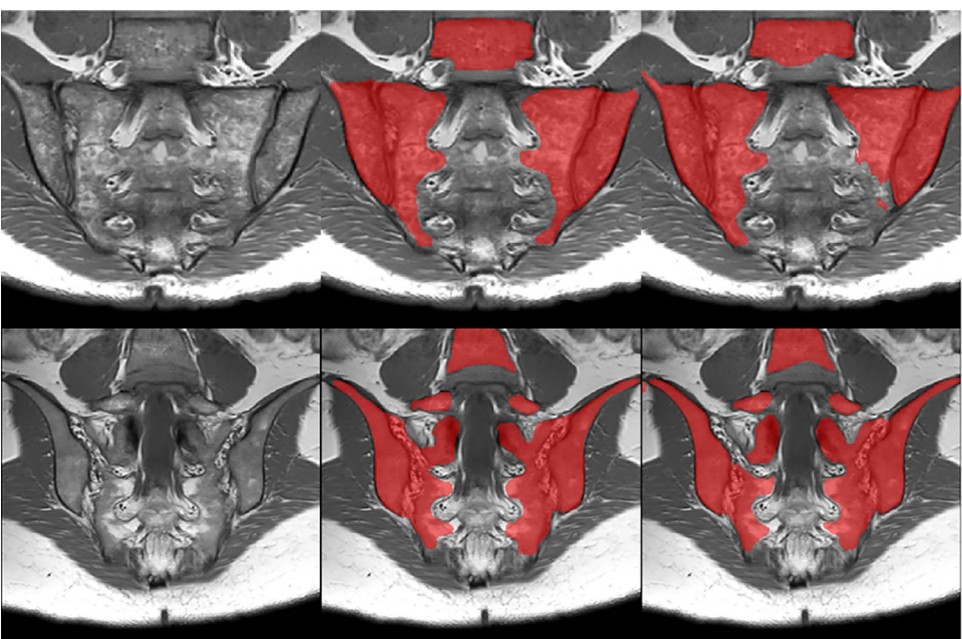

**Fig 2. Automatic segmentation of disease region: Demonstration of performance on examples from the test dataset.** The T1w images (left-hand column), reference standard (middle column) and model averaging ensemble prediction of disease region (right-hand column) are shown for illustrative slices from two subjects (top and bottom row).

times (for each of the four validation folds) into a training subset (150 image slices, 6 subjects) and validation subset (50 image slices, 2 subjects); this subdivision was performed at subject level to avoid any 'contamination' of the test dataset as a result of similarity individual subjects' image slices.

*Data pre-processing and augmentation.* To allow the same intensity scale between subjects and consistency in intensity levels of voxels representing the same tissue for each subject, images were normalized by three standard deviations of the image intensity distribution. Each pre-processed image (and the corresponding segmentation masks from the reference standard) underwent elastic deformation (https://github.com/gvtulder/elasticdeform/tree/v0.4.9), affine transformation (rotation, scaling, shearing) and random flip with 0.5 probability. To make the network robust against between-subject variations in the intensity level of bone voxels, intensities were raised to a random power after normalisation. All transformation parameters (rotation angle, scaling and shearing factors, power) were randomly sampled from uniform distributions of pre-defined ranges.

*Model training.* A convolutional neural network with two-dimensional U-Net architecture [11] was trained on mini-batches by optimizing binary cross entropy loss [13] using the Adam optimizer [14]. A publicly-available implementation in Pytorch was used (https://github.com/jvanvugt/pytorch-unet). The architecture included batch normalization to keep the distribution of convolution layers outputs fixed, allowing faster convergence [15]. The network was trained with pre-processed data augmented on the fly, which allowed a substantial increase of the diversity in the training samples. At each training epoch 350 augmentation steps were performed [16]. At each step, a random batch was selected from the available pool, augmented, and fed into the model. Data shuffling ensured that the same batch contained different image slices every epoch. Optimal hyper-parameters were identified through training with four-fold cross validation, specifically, (i) the number of epochs (60, 100), (ii) the number of resolution levels (2,4,6), where a level represents all feature maps between two max-pooling or two up-sampling operations [16] and (iii) convolution kernel size (3×3, 5×5). The batch size (four) and learning rate (0.001) were kept constant. The model parameters were initialized using the default Pytorch initialization scheme.

Once the optimal set of hyper-parameters was determined, the network was trained three times to reduce individual model's errors, using 200 image slices (8 subjects). Average prediction from three models was computed, then rounded, and performance of model averaging ensemble was evaluated.

*Model evaluation.* The performance of the model (averaging ensemble) for disease region segmentation was evaluated in two ways. First, to provide an evaluation in terms of the Dice coefficient (details in S1 File), performance was assessed against manual segmentation on the test dataset of two subjects (48 slices). Second, to assess the generalisability on the model on a variety of clinical cases, a further evaluation was performed by visual assessment (either satisfactory or not satisfactory) on a further 19 subjects (38 scans, 950 image slices), for which manual segmentations were not performed. Note that, to make use of all the available annotated data the model was re-trained three times using all 10 subjects, i.e. 248 slices, prior to the evaluation on the further 19 subjects. The visual assessment was performed by a postdoctoral researcher with two years of MRI experience who had received training in interpretation of the relevant anatomy by a radiologist; this radiologist also performed a slice-by-slice review of the segmentations in a subset of the cases to ensure that these anatomical assessments were accurate.

For the purposes of the subsequent assessment of the performance of the *complete workflow*, disease region segmentations (manual and automated) from 29 subjects were used. If the automated segmentation failed, the segmentations were manually corrected for use in the subsequent evaluation.

**2.5.4 Details of step (vi)–'Cleaning'.** The manual correction ('cleaning') procedure in Step vi was performed by two consultant radiologists, with over 25 and 7 years of musculoskeletal MRI experience respectively, as follows. Regions which were deemed non-inflammatory– for example due to the presence of vessels or artefact–were removed by readers based on morphology and anatomical location. Note that all images undergo this correction procedure, but the extent to which the images are actually corrected or 'cleaned' depends on the accuracy of the preliminary segmentation, as judged by the human observer.

To minimise subjectivity, lesions were either left in place or removed altogether, i.e., the boundaries of lesions were not modified, except when the posterior part of the joint or foramen were segmented along with a potential lesion. To facilitate the comparison with visual scoring of bone marrow oedema (see details in the following section), areas of inflammation located above the L4/5 disc and within the sacroiliac joint space were removed by the observers as part of the cleaning process, meaning that the cleaned segmentations contained only subchondral bone marrow oedema.

$T_1$-weighted images were used to assist readers in identification of anatomical structures and regions of increased fat content. The two readers discussed and agreed upon the procedure prior to manual correction.

## 2.6 Conventional visual scoring

To provide a comparator to $V_{HI}$, visual scoring was performed on the same STIR images as those used for the semiautomatic segmentation using the SPARCC BME system [2], by the same two consultant radiologists as performed the cleaning procedure. Images were read in random order on a dedicated research workstation where the reader was blinded to clinical diagnosis, treatment and all quantitative measurements, including $V_{HI}$. The presence of bone marrow oedema (BME) was evaluated in six consecutive slices, with the SIJs divided into eight quadrants. Each quadrant was scored for the presence/absence of BME (1 or 0) with an additional score of 1 if the BME in a quadrant was more than 10mm deep and a further additional score if the BME was at least as intense as the cerebrospinal fluid. A total score out of 72 was reached for SPARCC BME.

## 2.7 Performance assessment

**2.7.1 Comparison of inter-observer segmentation overlap—semiautomated pipeline versus purely manual segmentation.** To characterise the variability in manual segmentation and to establish a baseline segmentation performance, both radiologists performed two sets of purely manual segmentations in a subset of eight patients. This design allowed separation of the effects of inter- and intra-observer variability on segmentation performance, and meant that poor performance due to differences in opinion/expertise could be distinguished from intrinsic difficulties with performing the task consistently. The segmentations were temporally separated by one month to minimise any learning effect, and the eight subjects were selected to provide a range of inflammation severities. Having established the performance baseline, inflammation was again segmented in the same eight subjects by the same radiologists using the semiautomated workflow.

Inter-observer was compared between the semiautomated and purely manual segmentations in terms of Dice scores (further detail is provided in S1 File). To provide a further evaluation in terms of accuracy, we constructed a composite reference standard using a majority vote from both methods. Voxels which were deemed inflamed at least three times from two manual segmentation trials and two semiautomated segmentation trials were taken to be truly inflamed. The performance of the two methods was compared against this composite reference standard in terms of Dice scores.

**2.7.2 Comparison of interobserver agreement—$V_{HI}$ versus visual scoring.** Before proceeding to the agreement analysis, the relationship between $V_{HI}$ scores and visual scores was analysed graphically using scatterplots. To improve visualization of $V_{HI}$ measurements clustered at the lower end of the range, scatterplots were generated with the raw data and also following (i) data truncation to remove the highest $V_{HI}$ values and (ii) log(x+1) transformation to linearize the relationship between $V_{HI}$ and visual scores whilst ensuring that 0 values are unaltered after transformation (for ease of interpretation). The relationship between $V_{HI}$ and visual scores was evaluated with linear regression; slope and intercept values were reported with 95% confidence intervals.

Bland-Altman 95% limits of agreement (LoA) analysis was performed for both $V_{HI}$ and SPARCC scoring. Plots were generated using raw data, truncated data and log(x+1)-transformed data across the full dataset of 29 patients. The mean bias and 95% LoA were calculated and reported using the raw, non-transformed data for both $V_{HI}$ and visual scores. For the purposes of this analysis, only inflammation present in the subchondral bone was included in the final cleaned segmentations, in order to facilitate a more direct comparison with SPARCC scoring (which includes only subchondral bone marrow oedema).

**2.7.3 Responsiveness to biologic therapy—$V_{HI}$ versus visual scoring.** Changes in $V_{HI}$ and visual scores after treatment were visualized using spaghetti plots in which changes in both measurements for individual patients were depicted (Fig 9). Separate plots were generated for those patients who showed evidence of response to therapy using clinical criteria and for those patients who did not. To provide a numerical summary of the ability of $V_{HI}$ to capture response, we recorded the agreement between clinical response assessment and $V_{HI}$-based response assessment and between clinical response assessment and SPARCC-based response assessment. For the purposes of this analysis, any patient undergoing an improvement in $V_{HI}$/SPARCC was deemed to be a $V_{HI}$/SPARCC responder. Note that this is an imperfect assessment and an alternative would be to have a threshold for response based on the variance in the data, however, the latter is problematic when the distributions of the data are so different for $V_{HI}$ and SPARCC and risks creating an unfair threshold depending on the specific transformation used. The proposed approach, whereby any improvement is regarded as a response, is not regarded as a clinically meaningful threshold but rather as a useful simplification for the purpose of this analysis.

**2.7.4 Failure analysis.** Error analysis was performed for any scan in which the difference in $V_{HI}$ between observers was more than two standard deviations from 0. The analysis was conducted by one of the two consultant-level readers. Specifically, the scans and accompanying segmentation masks for each 'error case' were inspected to determine the reasons for discrepancy; errors were classified as anatomical (relating to whether observers agreed that hyperintense regions classified as subchondral or not, i.e. whether they were in a realistic anatomical location for inflammation to occur), morphological (relating to whether observers agreed that hyperintense regions were true oedema rather than vessels or other structures, i.e. whether the morphology was appropriate for an inflammatory lesion) or artefact-related (relating to whether hyperintense regions were deemed artefactual, i.e. due to a spurious region of high signal in the image that is erroneously introduced during the reconstruction of the images by the scanner).

# 3 Results

## 3.1 Overview

An example of normal bone segmentation is shown in Fig 1B, an example of disease region segmentation is shown in Figs 1C and 2, and an example of inflammation segmentation

(without the final cleaning step) is shown in Figs 1E and 3. The final cleaning step is highlighted in Fig 1F. The following subsections describe specific evaluations of the individual components of the workflow, and of the performance of the workflow as a whole for inflammation assessment.

### 3.2 Disease region segmentation performance

The model was successfully trained without evidence of overfitting. The learning curves for the training procedure (for different training subsets and validation folds) are shown in S1 Fig.

Assessing performance in terms of Dice scores (test dataset of 48 slices from two subjects), the model ensemble yielded a mean (range) Dice of 0.94 (0.85 to 0.98), indicating excellent segmentation performance. Examples of automatically segmented disease regions in the test dataset and the corresponding reference standard segmentations are shown in Fig 2.

Assessing performance qualitatively (19 further subjects), model performance was either perfect or subject to minor corrections for 16 subjects. The model failed in three subjects, each of whom was found to have abnormal bone marrow (high fat content or extensive sclerosis, which were not present in the training dataset). Examples of model failures are shown in S2 Fig.

### 3.3. Comparison of inter-observer segmentation overlap for final cleaned segmentation–semiautomated pipeline versus purely manual segmentation

The improvement in segmentation performance provided by the workflow is shown in terms of Dice scores in Fig 4.

For purely manual segmentation (without the workflow), Dice scores from the two observers' segmentation volumes ranged from 0.28 to 0.87. Intra- and inter-reader median Dice values were 0.63 and 0.69 for reader 1 and 2 and in the range 0.53–0.56, respectively.

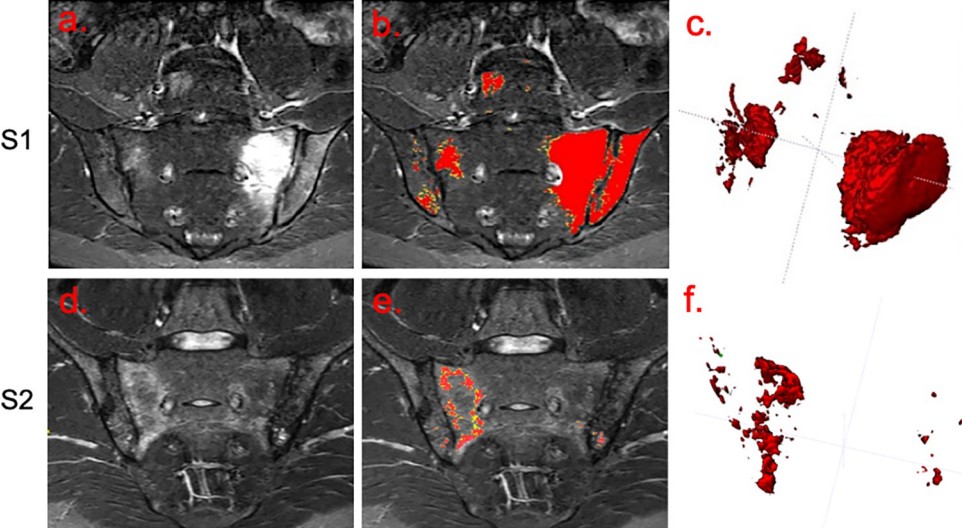

**Fig 3. Example outputs from human-machine workflow.** Scans for two subjects are shown (S1 and S2, top and bottom row respectively). The left-hand column shows the STIR images, the middle column shows the preliminary segmentations (not cleaned to provide a demonstration of the performance of the automated component) and the right column shows visual summaries of the disease volume. Note that inflammation at the periphery of segmented lesions is typically captured by the lower 'sensitive' threshold, shown in yellow, whereas the most hyperintense inflammation in the centre of lesions is captured by the higher, 'conservative', threshold, shown in red.

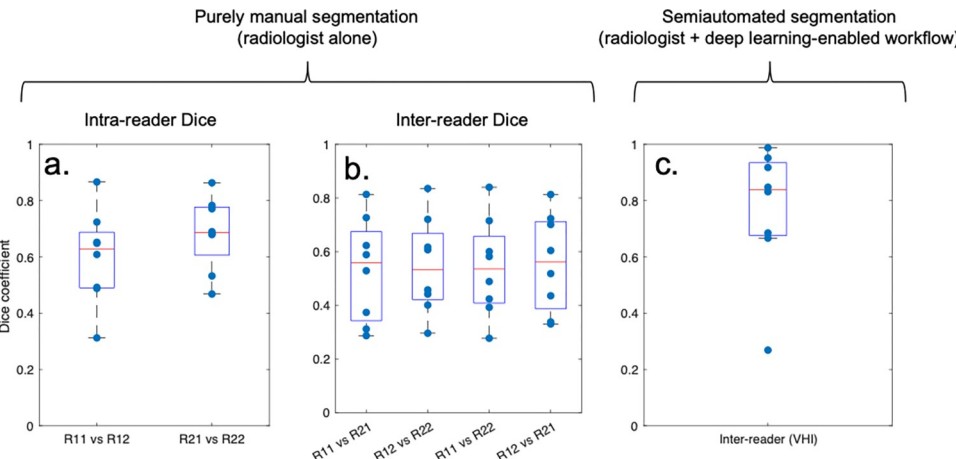

**Fig 4. Improvement in Dice scores for human-machine workflow compared to purely manual segmentation.** Dice scores are shown for individual patients for trials of segmentation of inflammation (volume comparison): (a) and (b) show results for purely manual segmentation and (c) shows results for corrected automatic (i.e. semiautomated) segmentations. (a) shows within-reader results, (b) shows between- reader results and (c) shows between-reader results. $R_{ij}$ stands for reader with the first subscript corresponding to the reader and the second to the segmentation trial. The figures show boxplots with individual datapoints superimposed; the red line represents the median dice.

Using the human-machine workflow, the median Dice scores improved to 0.84, representing an increase of 28–31% compared to pure manual segmentation. The inter-observer Dice for the two readers using the human-machine workflow was substantially higher than the overlap between each reader and the preliminary, non-cleaned segmentation (0.60 / 0.55 for Readers 1 and 2 respectively).

VHI scores from those obtained from the human-machine workflow also agreed closely with those obtained from a composite of all four manual segmentations (two from each reader) (Fig 5).

There was one outlier where the agreement was reduced for the semiautomated method; review of the images indicated that the disagreement mostly related to the presence of inflammation in the joint space, where blood vessels can be misinterpreted.

## 3.4 Comparison of interobserver agreement—$V_{HI}$ versus visual scoring

The relationship between $V_{HI}$ and visual scoring is shown in Fig 6. Note that $V_{HI}$ shows a nonlinear relationship with SPARCC scoring, reflecting the fact that SPARCC scoring gives binary scores for each quadrant and therefore effectively 'plateaus' at higher inflammation volumes. The relationship becomes approximately linear with logarithmic transformation.

Bland-Altman limits-of-agreement plots for $V_{HI}$ and visual SPARCC scoring are shown in Fig 7. The Bland-Altman LoA were +191 (-4119 to 4501) voxels over a range of 9.5 to 90081 for $V_{HI}$ and -1.5 (-14.8 to 11.8) voxels over a range of 0 to 47.5 for SPARCC scoring. Note that the limits are narrower for $V_{HI}$ than visual scoring relative to the range of mean values in the data, suggesting improved inter-observer agreement.

After logarithmic transformation, the widths of the LoA were similar for $V_{HI}$ and visual scoring relative to the range of values in the data, suggesting similar inter-observer agreement. However, note that the log-transformed data highlights greater disagreement in cases where the clinical burden of inflammation is small. This is natural because decisions on the presence / absence of inflammation are more difficult when only small regions are involved.

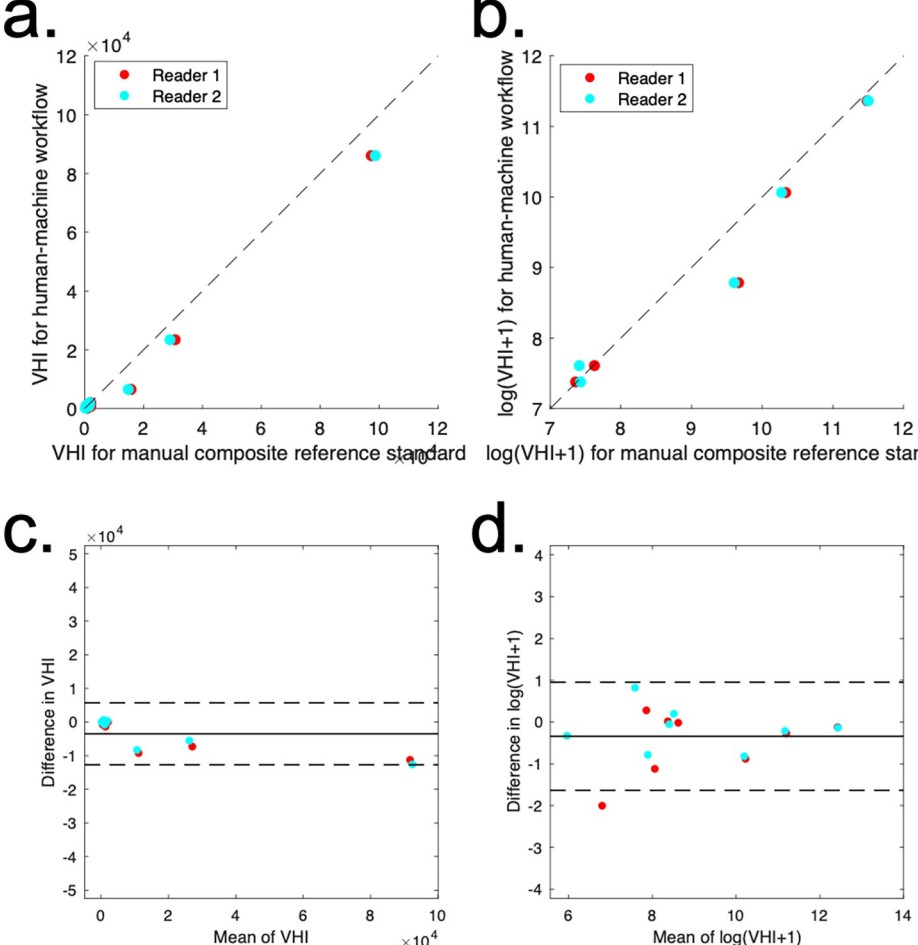

**Fig 5. Accuracy of human-machine workflow against a composite manual reference standard formed from all four human segmentations (two from each reader).** In the scatterplots (top row), the dotted line represents unity. In the Bland-Altman plots (bottom row) the mean bias (solid line) and 95% limits of agreement (dashed lines) are shown. Both plots show only a small bias and good precision relative to the range of $V_{HI}$ values present.

Furthermore, note that the difference in log-transformed scores indicates proportional disagreement, which can be large even when the absolute size of the differences is small.

### 3.5 Responsiveness to biologic therapy—$V_{HI}$ versus visual scoring

Examples of pre- and post-treatment scans and accompanying segmentations for a single subject are shown in Fig 8, and response plots are shown in Fig 9.

16/29 patients underwent a clinical response. Of the 16 clinical responders, 11/16 were also classified as responding by $V_{HI}$ and 12/16 were classified as responding by SPARCC scoring. Of the clinical non-responders (13/29), 4/13 were also classified as non-responding by $V_{HI}$ and 6/12 were classified as non-responding by SPARCC scoring.

$V_{HI}$ and clinical assessment agreed on response/non-response in 15/29 subjects. SPARCC scores and clinical assessment agreed on response/non-response in 18/29 subjects. SPARCC scores and $V_{HI}$ agreed on response/non-response in 19/29 subjects.

There was a significant linear relationship between the change in $V_{HI}$ and the change in SPARCC scores, with an estimated regression slope (95% CI) of 2408 (1230 to 3586) ($P = 0.0003$), an estimated intercept of 894 (-10760 to 12547) ($P = 0.88$) and $R^2 = 0.39$.

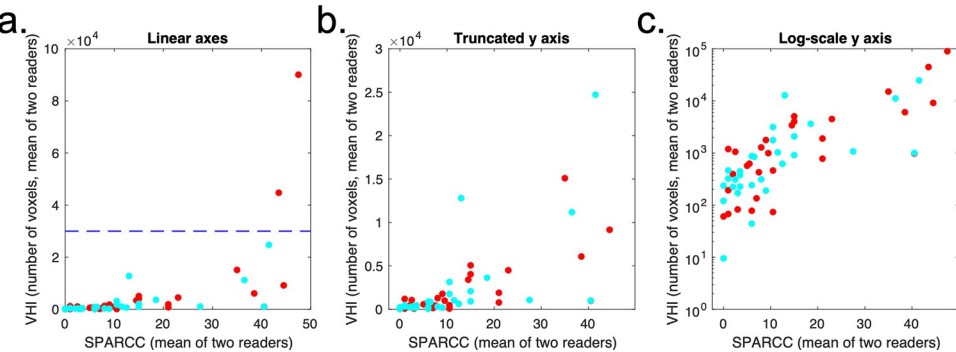

**Fig 6. Relationship between VHI and conventional visual scoring.** The raw voxel counts (left column), voxel counts with truncation of the y-axis (middle column) and voxel counts with log(x+1) transformation (right-hand column) are shown on scatterplot. The truncation point used to generate (b) is shown on (a) as a blue dotted line. Note that the relationship between SPARCC scoring and inflammation volume is nonlinear but becomes approximately linear with log(x+1) transformation. Red points refer to pre-treatment scans, and blue points to post-treatment scans. R1 and R2 refer to reader 1 and reader 2.

## 3.6 Failure analysis

Discrepancies between the two observers for the semiautomated segmentation were identified in 3/58 cases. The images from these cases are shown in S3 Fig. Inspection of these images revealed that disagreement was 'anatomical' in all two cases (i.e. relating to the location of hyperintensity) and 'artefactual' (i.e. relating to whether hyperintense regions were deemed artefactual) in one case; there were no instances of morphological (e.g. relating to whether hyperintense regions were classified as oedema rather than vessels) disagreement. Specifically, in the two cases of 'anatomical' disagreement, the observers disagreed due to the presence of hyperintense bone in the posterior ilium, which was attributed to inflammation by one observer but to anatomical variation by the other. In the one case of 'artefactual' disagreement, there was diffuse and mild hyperintensity in subchondral bone which was deemed inflammatory by one reader but normal by the other; this was a post-treatment scan in a patient with extensive inflammation that had improved after treatment–i.e. the difficulty in this case related to the identification of resolving inflammation.

## 4 Discussion

At present, there is no imaging biomarker of inflammation that is used widely in clinical practice, and image interpretation is performed in a qualitative fashion, introducing substantial subjectivity. Although deep learning is a natural approach to addressing this problem, purely automated solutions require large training datasets and may not be sufficiently accurate or trustworthy for use in clinical practice when only limited data are available for training. Here, to address these issues, we propose a hybrid 'human-machine' workflow that aims to combine deep learning-based segmentation with human oversight. The output of this workflow defines a quantitative imaging biomarker known as the volume of hyperintense inflammation ($V_{HI}$).

By first using a U-net—the current state-of-the-art approach for medical image segmentation [11, 17]—to recognize potentially inflamed bone and then segmenting areas within this using thresholding, our approach has the advantages that (i) the disease region segmentation is relatively trivial and can be achieved with a relatively small dataset, (ii) the segmentation of inflammation is transparent, easily-understood and objective, removing the need for subjective intensity-based judgements to be made by radiologists, and (iii) these segmentations can easily be propagated onto images from other sequences, and could easily be applied to

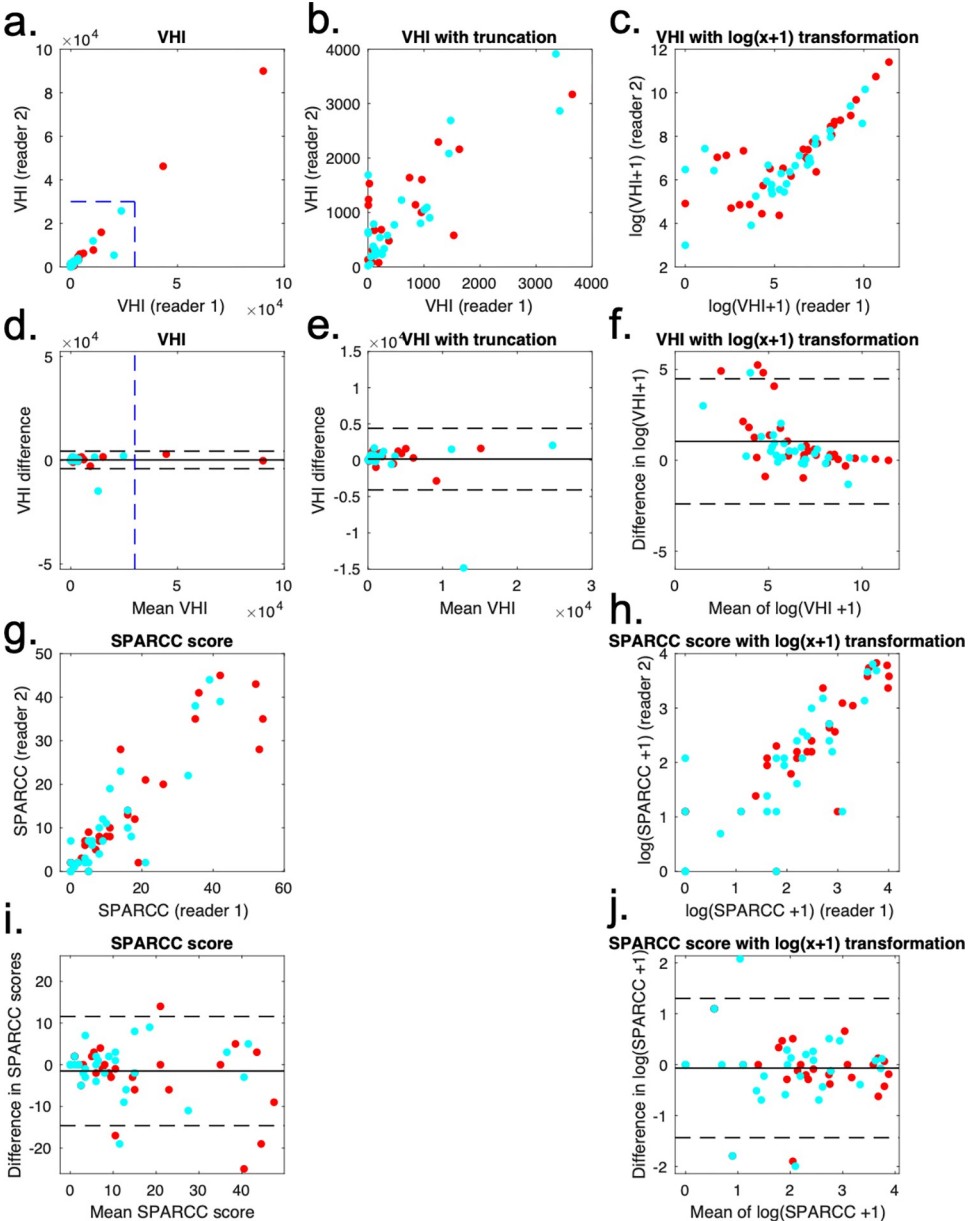

**Fig 7. Improvement in inter-observer agreement for V_{HI}.** Bland-Altmans plots for the two observers' scores are shown for VHI (top half) and SPARCC scores (bottom half). Red points refer to pre-treatment scans, and blue points to post-treatment scans. For VHI, the raw voxel counts (left column), voxel counts with truncation of the axes (middle column) and voxel counts with log(x+1) transformation (right-hand column) are shown on scatterplot (top row) and Bland-Altman plots (second row). The truncation points used to generate the middle column figures are shown as blue dotted lines on the plots in the left column. For SPARCC scores, the raw scores (left column) and log(x+1) transformed scores (right column) are shown on scatterplots (third row) and Bland-Altman plots (fourth row).

quantitative MRI. Furthermore, the semiautomated nature of the workflow means that the burden placed on the radiologist is minimised (since cleaning a sensitive segmentation such as this requires removal of only 'chunks' of oversegmented tissue such as artefacts and vessels, a quick and simple process), making it amenable to use within a standard radiological workflow. Our approach is broadly similar to how semiautomated segmentation is already used for a widely-used and impactful technique known as coronary calcium scoring, where an initial

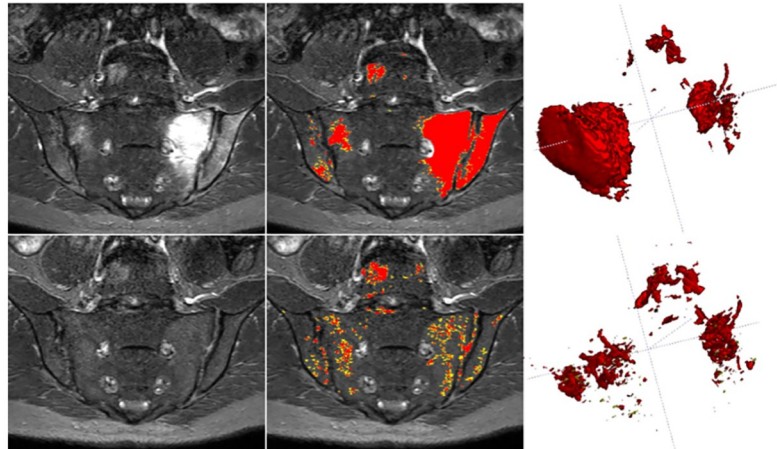

**Fig 8. Example of response to biologic therapy.** Pre- and post-treatment scans for a single subject are shown (top and bottom row respectively). The left-hand column shows the STIR images, the middle column shows the preliminary segmentations (not cleaned to provide a demonstration of the performance of the automated component) and the right column shows visual summaries of the disease volume. Regions of acute inflammation showed a reduction in extent and intensity after treatment, although there is a persistent, slight hyperintensity compared to the normal interforaminal bone, which manifests as an increase in the proportion of inflammation captured by the lower (yellow) of the two thresholds (see Details of Step (ii)–Thresholding within disease regions).

segmentation is generated by thresholding and then a radiologist adjusts the segmentation to retain only relevant regions [18–22]. From a clinical perspective, $V_{HI}$ measurements provides similar information to SPARCC scoring (which is confined to the research setting) yet avoids the need for subjective and laborious visual assessment of image intensity. $V_{HI}$ measurements should be simpler to interpret for clinicians than qualitative reports, and the accompanying segmentations provide a visual illustration of disease burden which is easy to understand for clinicians and patients.

The key results of our study are as follows. Firstly, the semiautomated workflow produced a marked improvement in inflammation segmentation performance compared to the purely manual approach in terms of inter-observer agreement. Second, $V_{HI}$ measurements show similar or better inter-observer agreement than visual scoring, although direct comparison is difficult due to the differences in the metrics' distributions: comparison on the non-log-transformed data suggests superior performance for $V_{HI}$ and comparison on the transformed data suggests similar performance. The former may be more representative of performance in clinical practice, where absolute differences are more relevant than proportional differences. Thirdly, $V_{HI}$ measurements show a nonlinear, approximately exponential relationship with visual scoring, which becomes approximately linear with logarithmic transformation. This result may reflect the fact that $V_{HI}$ can capture the full burden of inflammation present in the subchondral bone whereas visual scoring is limited to binary assessments for each quadrant of the joint, and thus does not distinguish between areas of inflammation of different sizes within a quadrant. From a clinical perspective, a technique with a greater dynamic range may be able to better stratify patients by inflammation burden and better capture changes in inflammation severity with treatment, even when inflammation does not completely resolve (for example, when performing early response assessments). Fourthly, $V_{HI}$ and visual scoring provide broadly similar response assessments, although neither metric agrees closely with clinical response assessments. The latter point probably reflects the complex, multifactorial nature of pain and the fact that this is not solely due to inflammation.

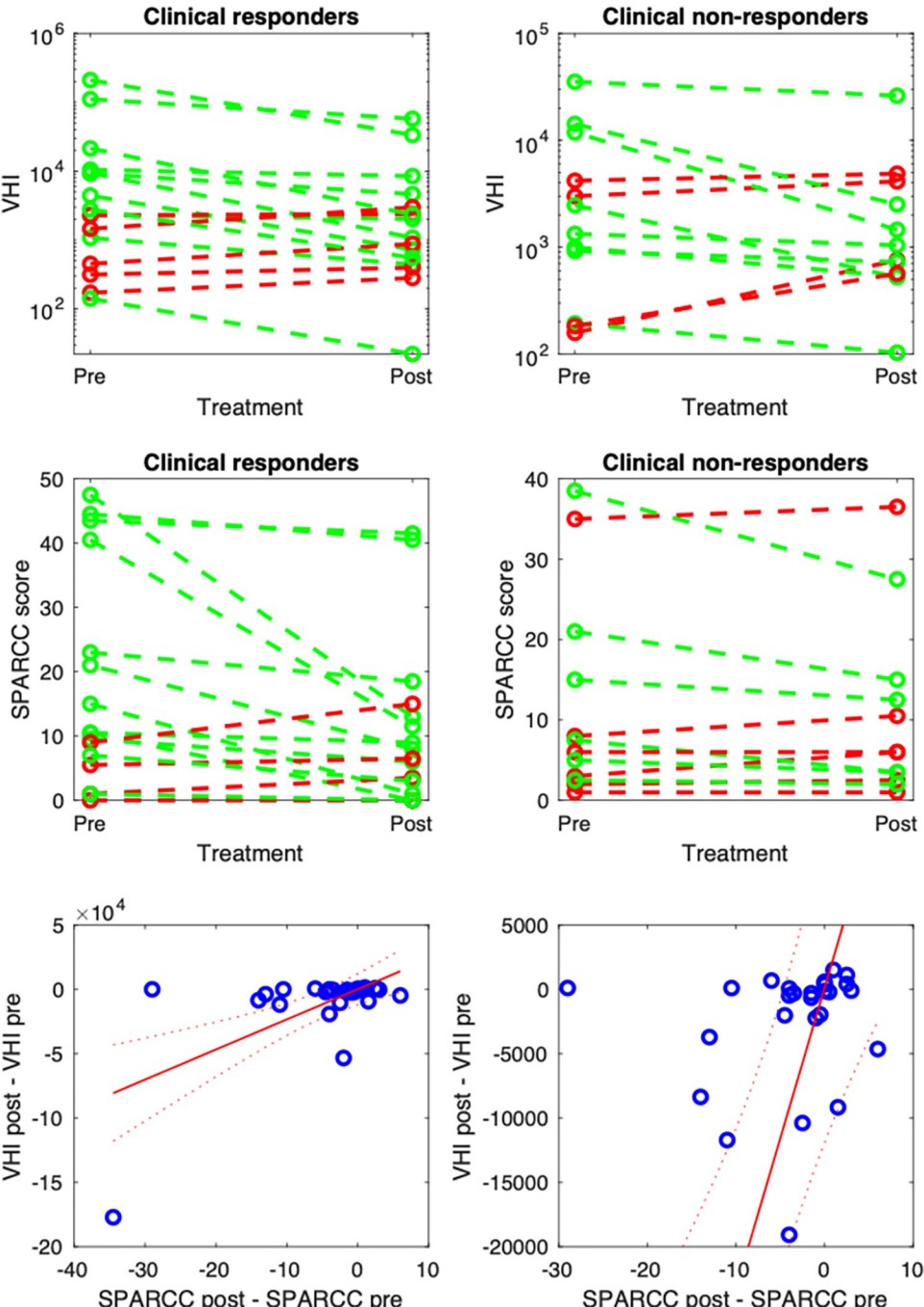

**Fig 9. Spaghetti plot for V<sub>HI</sub> and SPARCC scores on pre- and post-treatment scans.** Subjects with improving inflammation (based on imaging assessments by either V<sub>HI</sub> or SPARCC score) are shown in green; subjects with worsening inflammation are shown in red. The first row shows results for VHI and the second shows results for SPARCC. The third row shows a scatterplot of VHI changes against SPARCC changes for individual subjects, using the full data range (bottom left) and truncated y-axes (bottom right). The linear regression line has identical parameters for the two plots on the bottom row.

Several previous studies have also investigated the use of threshold-based methods for quantifying inflammation [23, 24]. However, these studies relied on manual segmentation to identify an optimal threshold, whereas our data suggest that using manual segmentation as a "gold standard" is problematic and may lead to inconsistent interpretation especially in cases when inflammation is subtle or precise lesion boundary cannot be identified. To highlight this point, a recent study aiming to demonstrate the feasibility of fully-automated segmentation of BME [25] revised the threshold value developed in earlier work [23], finding an optimal threshold of 1 compared to 1.5 in the prior study. Clearly, a threshold which depends on reference standard provided by human observers is not desirable. In contrast, the approach proposed in this work removes the need for intensity-based judgements to be made by the observer. The use of an intensity-based threshold derived from normal marrow means that the choice of voxels is primarily influenced by the physical properties of the tissue, specifically, the extent to which the intensity in each voxel deviates from the intensity observed in normal marrow. The normal bone region effectively serves as a reference region and means that the judgment around which voxels are hyperintense is tailored to each individual and each scan.

Importantly, $V_{HI}$ measurements should be simpler to interpret for clinicians than qualitative reports, which vary in style and length between radiologists and depend on expertise and opinion. The segmentation masks generated by the workflow could be displayed together with the $V_{HI}$ measurement, providing a visual illustration of disease burden which is easy to understand for clinicians and patients. Visual illustrations could make disease activity assessments easier to understand for patients and help them to feel more in control of their disease and care.

## 4.1 Limitations

This study has several limitations. First, the network was trained on a relatively small dataset, and produced errors in cases which were atypical. However, the intention of this study is not to provide a definitive final algorithm, but to demonstrate the potential of the proposed deep-learning enabled workflow. The improvement in performance showed by our data suggests that further development, which might include network training on a larger, multisite dataset (thus introducing greater robustness to atypical cases), is warranted. Secondly, the performance of the method is fundamentally limited by the acquisition modality, which in this case was STIR imaging. Although widely used, STIR imaging has a number of limitations including its relatively poor signal-to-noise ratio and the potential for inadequate fat suppression. However, a strength of our approach is that it can easily be applied to other imaging modalities, including quantitative imaging, since the disease region masks can easily be propagated to other modalities. This is a substantial advantage comparing to requiring a network that is directly trained to identify inflammation on specific sequences. Thirdly, although we showed a substantial performance improvement for the human-machine workflow compared to qualitative assessment, this is essentially an agreement study and we do not have a true 'gold standard' for accuracy assessment. Although obtaining a gold standard is challenging (obtaining histology, for example, is limited by ethical constraints), one potential would be to create a composite or consensus reference standard using a large number of radiologists, and then assess the performance of further radiologists (with and without machine assistance, and potentially with varying levels of experience) against this standard. Ideally, this dataset would include scans from multiple centres to enable an assessment of generalisability of algorithms. However, at present we are not aware of such a dataset; the study design used here is a practical way to assess the feasibility of this approach. Finally, our results do indicate that subjective judgements made in the cleaning step of the workflow have a substantial impact on $V_{HI}$

measurements, although the use of the semiautomated workflow improved agreement between observers. Further research could therefore focus on greater automation of the method, including automatic removal of vessels and image artefacts, further reducing subjectivity. One approach would be to train the U-net to directly identify inflammation from the STIR images, however, this is a more challenging task and this approach could therefore be less generalisable to unseen data (e.g. from other scanners or sites), as well as less transparent. An alternative would be to train a separate algorithm to perform the cleaning step. We suggest that future research could focus on examining the interplay between the degree of automation, the time taken for segmentation and the degree of accuracy and precision that can be achieved.

## 4.2 Conclusion

We propose a workflow for segmentation of inflammation incorporating both deep learning and human input. The output of this workflow, the volume of hyperintense inflammation ($V_{HI}$), provides a precise assessment of inflammation with superior performance to visual scoring by trained expert radiologists. The proposed human-machine workflow for $V_{HI}$ measurement offers a mechanism to improve the consistency of radiological assessment of inflammation, and a biomarker of inflammation load to guide treatment decisions in spondyloarthritis. It could also be a useful exemplar of human-machine cooperation more broadly.

## Supporting information

**S1 File. Evaluation metrics.**
(DOCX)

**S1 Fig.** Mean area overlap (Dice score) vs training epoch for different training data subsets (a) and validation folds (b). Each point represents Dice score averaged over (i) classes (foreground & background), (ii) samples in a mini batch and (iii) 350 augmentation steps. Area overlap from pair-wise comparison of reference standard and rounded prediction on the test data from models averaging ensemble (three runs using all training data, 200 T1W image slices) (c).
(TIF)

**S2 Fig. Examples of model failure in disease region segmentation.** T1w image slices in oblique coronal plane (top) for three subjects with super-imposed models averaging ensemble rounded prediction (bottom). Subjects exhibit very abnormal bone, comprising either high fat content (left, middle) or sclerosis (right), leading to areas of 'missing' bone within the segmentations.
(TIF)

**S3 Fig. Examples of discrepancies between readers for the cleaning step.** The three discrepancies are denoted D1-D3 and shown on separate rows; for each, the STIR image (left column) and segmentations for the two readers (middle and right column) are shown. The green and red segmentations correspond to the higher and lower segmentation thresholds. In two cases (D1, D2), the disagreement was 'anatomical' and related to the presence of hyperintensity in the posterior ilium, which can be attributed to either inflammation or variations in normal bone composition. In one case (D3) the disagreement was 'artefactual' and related to the presence of faint, diffuse hyperintensity in the potentially-inflamed subchondral bone region, which was deemed entirely inflammatory by one reader and partly artefactual by the other.
(TIF)

## Author Contributions

**Conceptualization:** Carolyna Hepburn, Alexis Jones, Alan Bainbridge, Hui Zhang, Margaret A. Hall-Craggs, Timothy J. P. Bray.

**Data curation:** Carolyna Hepburn, Alexis Jones, Alan Bainbridge, Timothy J. P. Bray.

**Formal analysis:** Carolyna Hepburn, Timothy J. P. Bray.

**Funding acquisition:** Margaret A. Hall-Craggs, Timothy J. P. Bray.

**Investigation:** Alan Bainbridge, Coziana Ciurtin, Hui Zhang, Timothy J. P. Bray.

**Methodology:** Carolyna Hepburn, Juan Eugenio Iglesias, Timothy J. P. Bray.

**Project administration:** Alexis Jones, Juan Eugenio Iglesias, Hui Zhang, Margaret A. Hall-Craggs, Timothy J. P. Bray.

**Resources:** Alan Bainbridge, Coziana Ciurtin, Hui Zhang.

**Software:** Carolyna Hepburn, Juan Eugenio Iglesias, Timothy J. P. Bray.

**Supervision:** Juan Eugenio Iglesias, Margaret A. Hall-Craggs, Timothy J. P. Bray.

**Validation:** Timothy J. P. Bray.

**Visualization:** Carolyna Hepburn, Timothy J. P. Bray.

**Writing – original draft:** Carolyna Hepburn, Alexis Jones, Alan Bainbridge, Coziana Ciurtin, Margaret A. Hall-Craggs, Timothy J. P. Bray.

**Writing – review & editing:** Hui Zhang, Margaret A. Hall-Craggs, Timothy J. P. Bray.

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
