## [Decision Letter · Decision Letter 0]

12 Jan 2023

PONE-D-22-24520Volume of hyperintense inflammation (VHI): a deep learning-enabled quantitative imaging biomarker of inflammation load in spondyloarthritisPLOS ONE

Dear Dr. Bray,

Thank you for submitting your manuscript to PLOS ONE. After careful consideration, we feel that it has merit but does not fully meet PLOS ONE’s publication criteria as it currently stands. Therefore, we invite you to submit a revised version of the manuscript that addresses the points raised during the review process.

In particular, I agree with the reviewers' comments about the need for some explanation throughout the manuscript.

Overall, I believe the presented work could be accepted after minor revision.

We look forward to receiving your revised manuscript.

Kind regards,

Paolo Cazzaniga

Academic Editor

PLOS ONE

Journal Requirements:

This work was funded by Action Medical Research, the Humanimal Trust and The Albert Gubay Foundation. TJPB is supported by an NIHR Clinical Lectureship (CL-2019-18-001). MHC is supported by the National Institute for Health Research (NIHR) Biomedical Research Centre (BRC). This work was undertaken at UCLH/UCL, which receives funding from the UK Department of Health’s the NIHR BRC funding scheme. The views expressed in this publication are those of the authors and not necessarily those of the UK Department of Health. 

No competing interests to declare. 

TJPB is supported by an NIHR Clinical Lectureship (CL-2019-18-001). MHC is supported by the National Institute for Health Research (NIHR) Biomedical Research Centre (BRC). This work was undertaken at UCLH/UCL, which receives funding from the UK Department of Health’s the NIHR BRC funding scheme. The views expressed in this publication are those of the authors and not necessarily those of the UK Department of Health. 

However, funding information should not appear in the Acknowledgments section or other areas of your manuscript. We will only publish funding information present in the Funding Statement section of the online submission form. 

This work was funded by Action Medical Research, the Humanimal Trust and The Albert Gubay Foundation. TJPB is supported by an NIHR Clinical Lectureship (CL-2019-18-001). MHC is supported by the National Institute for Health Research (NIHR) Biomedical Research Centre (BRC). This work was undertaken at UCLH/UCL, which receives funding from the UK Department of Health’s the NIHR BRC funding scheme. The views expressed in this publication are those of the authors and not necessarily those of the UK Department of Health. 

Reviewers' comments:

Reviewer's Responses to Questions

**Comments to the Author**

1. Is the manuscript technically sound, and do the data support the conclusions?

Reviewer #1: Yes

Reviewer #2: Yes

2. Has the statistical analysis been performed appropriately and rigorously? 

Reviewer #1: Yes

Reviewer #2: Yes

3. Have the authors made all data underlying the findings in their manuscript fully available?

Reviewer #1: Yes

Reviewer #2: Yes

4. Is the manuscript presented in an intelligible fashion and written in standard English?

Reviewer #1: Yes

Reviewer #2: Yes

5. Review Comments to the Author

Reviewer #1: The authors present a semi-automated workflow for quantitative imaging and diagnosis in STIR images. The workflow uses a deep neural-net (UNET) to segment regions of inflammation, while also relying on manual steps for image clean-up. The automatically identified regions of inflammation can then be used to create a quantitative assessment of the image, via volume of hyperintense inflammation (VHI), and potentially increase the diagnostic power, especially when compared to standard visual assessment approaches (e.g. SPARCC).

Overall, this was a well written paper with a grounded approach and corresponding discussion which underscored the potential role of AI-based image processing and quantitative imaging biomarkers in a clinical setting. The methods used in this study were sound and explained with adequate detail, and the various diagnostic criteria and scoring methodologies were clearly outlined for the reader. The discussion highlights the main conclusions of the various performance/comparison testing, and the Limitations section appropriately discusses the potential pitfalls of over-reaching conclusions from this limited study.

I recommend this study for publication pending revisions based on the comments/questions below.

1. Do all images go through step vi in the work flow (i.e. the manual cleaning process)? Or are only images which ‘fail’ segmentation put through manual cleaning? Could the authors please specify these details. How are images deemed ‘failures’?

2. Additional discussion of the necessity/time burden for the manual cleaning step of the workflow is warranted. Can this step also be automated? If so, what is the approach suggested by the authors? If not, how do the authors envision this manual step fitting into the workflow in a real clinical environment?

3. The last discussion point in Section 4.1 reads: “… our results do indicate the subjective judgments made in the cleaning step of the workflow have a substantial impact on the Vhi measurements.” Unfortunately, this last point potentially undermines the claims in the study. Could the authors please estimate the impact or provide a performance comparison with and without the manual clean-up step?

4. Could the authors also include in the Discussion how they envision creating future data sets to improve their algorithm? Importantly, what is the best way to define the ‘gold standard’ segmentation or diagnosis? Should machine learning approaches be using consensus results from trained physicians, or is the current approach adequate?

5. Further discussion in section 4.1 on the limitations of the defined ‘gold standard’ is warranted. Are there better approaches to creating a standard image, i.e. through consensus of board-certified clinicians?

6. In the discussion, 2nd paragraph: “ comparison on the non-transformed data…” Could the authors please specify the ‘transformation’ they are referring to? Is this purely the log transformation?

7. Figure 5 caption should define R1 and R2.

8. Figure 7, the bottom left-most plot appears to have mislabeled axes. Should the axes be the difference between scores, not a comparison between readers? Additionally, I recommend labeling these plots a, b, c etc. to help guide the reader.

9. Figure 9 should be referenced in section 2.7.3

Reviewer #2: In this article, Hepburn et al present a workflow to analyze MRI images that enables precise quantification of VHI (volume of hypertense inflammation), through a combination of a CNN, intensity based thresholding, and a manual cleaning step. This work compares the performance of the semiautomated image analysis pipeline with manual SPARCC scoring, and convincingly shows that the presented approach is able to extract the VHI from STIR sequences. By analyzing a single image, segmentation can be propagated to the remainder slices, thus simplifying VHI extraction in comparison to individual image segmentation. By defining a ‘normal bone’ region, the workflow enables segmentation of MRI images that differ in intensities, textures, and plausibly, acquisition settings. This work highlights the usefulness of CNNs and computer vision not only to facilitate image diagnostics, but also to make the process more rigorous and accurate. It appears that VHI is not a common metric to assess disease state or clinical outcomes, and thus this work aims to both develop a pipeline for accurate VHI quantification, as well as propose VHI as a feature to be relied on as a clinically-relevant metric. The latter is not fully validated in this work. However, the presented work, although not fully optimized, nicely shows how computer vision can be harnessed for medical diagnostics. I believe this work would be of interest to a broad community. I have a few items I believe should be addressed prior to publication:

1. Can the authors provide some background for non-experts on the rationale or background to use VHI as a biomarker for sponduloarthritis?

2. Can the authors comment on whether the pipeline would be robust to MRIS acquired with different acquisition settings?

3. It would appear that the lower intensity limit (L_lower) can be simply calculated by knowing the value of IQR. The need for an iterative search is unclear

4. How common is it for patients to have ‘abnormal bone marrow’, that would render the pipeline unusable?

5. Fig 4 and 5 seem a bit redundant. If there is a standard reference that contains information from multiple readers, but the readers don’t tend to agree. This point could suggest the standard reference is not as reliable as possible. Could a third observer be included?

6. Although it is clear that the relationship between VHI and SPARCC is improved with a logarithmic transformation, it is unclear why this is so, or why this is justifiable. Can the authors expand on this choice?

7. Fig 9: linear relationship between VHI and SPARCC change, I do not think the slope or intercept values are as informative as the fit, can the authors provide the R2 as well?

8. The failure analysis lacks evidence. Determining why two of the samples were deemed ‘anatomical’ failures and the other one was ‘artefactual’ is not clear. This could be because it requires expertise on this type of images, in which case further explanation is warranted. If there is no data to support these claims, I suggest they are left to the reader’s interpretation. In addition, what is the significance of failures being anatomical or artefactual?

9. Is there anything potentially useful included in SPARCC scoring that VHI does not take into account?

Minor comments

10. I recommend defining acronyms, the manuscript is loaded with MRI / radiology jargon and acronyms and can be hard to follow.

11. In 3.3, is this referring to VHI or disease region?

12. When classifying patients as responders / non-responders, how was this performed when using VHI or SPARCC scoring? Was a specific % change deemed necessary? If so, how was this determined?

13. Fig 4c. I believe the caption is incorrect

6. PLOS authors have the option to publish the peer review history of their article (what does this mean?). If published, this will include your full peer review and any attached files.

Reviewer #1: No

Reviewer #2: No

---

## [Author Response · Author response to Decision Letter 0]

5 Mar 2023

See attached response to reviewers.

---

## [Decision Letter · Decision Letter 1]

3 Apr 2023

Volume of hyperintense inflammation (VHI): a quantitative imaging biomarker of inflammation load in spondyloarthritis, enabling by human-machine cooperation

PONE-D-22-24520R1

Dear Dr. Bray,

We’re pleased to inform you that your manuscript has been judged scientifically suitable for publication and will be formally accepted for publication once it meets all outstanding technical requirements.

Kind regards,

Paolo Cazzaniga

Academic Editor

PLOS ONE

Additional Editor Comments (optional):

Reviewers' comments:

Reviewer's Responses to Questions

**Comments to the Author**

1. If the authors have adequately addressed your comments raised in a previous round of review and you feel that this manuscript is now acceptable for publication, you may indicate that here to bypass the “Comments to the Author” section, enter your conflict of interest statement in the “Confidential to Editor” section, and submit your "Accept" recommendation.

Reviewer #2: All comments have been addressed

Reviewer #3: All comments have been addressed

2. Is the manuscript technically sound, and do the data support the conclusions?

Reviewer #2: Yes

Reviewer #3: Yes

3. Has the statistical analysis been performed appropriately and rigorously? 

Reviewer #2: Yes

Reviewer #3: Yes

4. Have the authors made all data underlying the findings in their manuscript fully available?

Reviewer #2: Yes

Reviewer #3: Yes

5. Is the manuscript presented in an intelligible fashion and written in standard English?

Reviewer #2: Yes

Reviewer #3: Yes

6. Review Comments to the Author

Reviewer #2: The authors have provided adequate explanation to my questions, and have addressed all concerns in great detail. I think the new arrangement of data presented in Figure 4 and 5 makes the paper stronger, and I appreciate the authors taking this into consideration.

Reviewer #3: The authors scanned and replied positively to all my queries. They also kindly examined and motivated well all their answers.

7. PLOS authors have the option to publish the peer review history of their article (what does this mean?). If published, this will include your full peer review and any attached files.

Reviewer #2: No

Reviewer #3: No

---

## [Editor Report · Acceptance letter]

11 Apr 2023

PONE-D-22-24520R1 

Volume of hyperintense inflammation (V_HI_): a quantitative imaging biomarker of inflammation load in spondyloarthritis, enabled by human-machine cooperation 

Dear Dr. Bray:

I'm pleased to inform you that your manuscript has been deemed suitable for publication in PLOS ONE. Congratulations! Your manuscript is now with our production department. 

Kind regards, 

on behalf of

Dr. Paolo Cazzaniga 

Academic Editor

PLOS ONE